# Abundance and Compositions of B-Vitamin-Producing Microbes in the Mammalian Gut Vary Based on Feeding Strategies

Hisham S. Alrubaye,[a] Kevin D. Kohl[a]

[a]Department of Biological Sciences, University of Pittsburgh, Pittsburgh, Pennsylvania, USA

**ABSTRACT** Mammals maintain close associations with gut microbes that provide numerous nutritional benefits, including vitamin synthesis. While most mammals obtain sufficient vitamins from their diets, deficiencies in various B vitamins (biotin, cobalamin, riboflavin, thiamine, etc.) are reported in captive animals. Biomedical and agricultural research has shown that gut microbes are capable of synthesizing B vitamins and assisting with host vitamin homeostasis. However, we have a poor understanding of distribution and abundance of B-vitamin synthesis across mammalian hosts. Here, we leveraged a publicly available metagenomic data set from 39 mammalian species and used MG-RAST to compare the abundance and composition of B-vitamin-synthesizing microbes across mammalian feeding strategies. We predicted that herbivores would have the highest abundance of genes associated with vitamin synthesis, as plant material is often low in B vitamins. However, this hypothesis was not supported. Instead, we found that relative abundances of genes associated with cobalamin and thiamine synthesis were significantly enriched in carnivorous mammals. The taxonomic community structure of microbes predicted to be involved in B-vitamin synthesis also varied significantly based on host feeding strategy. For example, the genus *Acinetobacter* primarily contributed to predicted biotin synthesis in carnivores but was not predicted to contribute to biotin synthesis in herbivores or omnivores. Given that B vitamins cannot be stored within the body, we hypothesize that microbial synthesis of B vitamins could be important for wild carnivores that regularly experience periods of fasting. Overall, these results shed light on the distribution and abundance of microbial B-vitamin synthesis across mammalian groups, with potential implications for captive animals.

**IMPORTANCE** Microbial communities offer numerous physiological services to their hosts, but we still have a poor understanding of how these functions are structured across mammalian species. Specifically, our understanding of processes of vitamin synthesis across animals is severely limited. Here, we compared the abundance of genes associated with the synthesis of B vitamins and the taxonomic composition of the microbes containing these genes. We found that herbivores, omnivores, and carnivores harbor distinct communities of microbes that putatively conduct vitamin synthesis. Additionally, carnivores exhibited the highest abundance of genes associated with synthesis of specific B vitamins, cobalamin and thiamine. These data uncover the potential importance of microbes in the vitamin homeostasis of various mammals, especially carnivorous mammals. These findings have implications for understanding the microbial interactions that contribute to the nutritional requirements of animals held in captivity.

**KEYWORDS** gut microbiome, vitamin biosynthesis

The mammalian gut microbiota is a complex community of bacteria, fungi, archaea, and viruses that supplies the host with numerous services, including the production of vital nutrients, such as amino acids, fatty acids, and vitamins (1, 2). Each mammalian species harbors a relatively distinct microbial community (3), and there is

Address correspondence to Kevin D. Kohl, kevin.d.kohl@gmail.com.

The abundance and compositions of B-vitamin-producing microbes in the mammalian gut vary based on feeding strategies, with higher abundances in carnivorous mammals.

interest in understanding the factors that drive the structure and function of these communities. For example, factors such as diet (4, 5), evolutionary history (6), and geography (7, 8) are known to influence aspects of microbial community structure. However, our understanding of how specialized microbial functions vary across mammalian species remains more limited.

Diet, in particular, has strong influences on the microbial community structure (9, 10) and function (11) of the mammalian gut microbiome. For example, carnivores and herbivores have differences in metabolic features encoded in their microbiomes; the microbiomes of carnivorous mammals are specialized for protein degradation, while the microbiota of herbivorous mammals are enriched in genes associated with synthesizing essential amino acids (11). Additionally, the gut anatomy of herbivores (foregut fermentation versus hindgut fermentation) can influence the community structure of the microbiome recovered in feces (10), though differences in microbial functions between these digestive strategies have not been studied. Overall, the relationship between host dietary strategies and nutritional functions other than fiber fermentation has been largely overlooked (12), and microbiome work that has been conducted in mammals has largely focused on agricultural and laboratory models (13).

Vitamin synthesis is an important metabolic function provided by the gut microbiome; however, we have a poor understanding of these processes. Vitamins are organic molecules that play vital roles as coenzymes in the metabolism of fat, protein, and carbohydrates and are essential for maintaining optimal health of organisms (2, 14). Here, we specifically focus on B vitamins, which can be divided into 8 types (biotin, cobalamin, folate, niacin, pantothenate, pyridoxine, riboflavin, and thiamine), all of which can be synthesized by gut microbes (2, 15) and absorbed in the hindgut (16). In insects, microbial synthesis of B vitamins contributes significantly to the health and development of insect hosts (17). For example, in bedbugs (*Cimex lectularius*), microbial synthesis of B vitamins aids hosts in overcoming the low B-vitamin content of blood and is required for host development (18). Additionally, in tsetse flies (*Glossina morsitans*), microbial production of vitamin $B_6$ (pyridoxine) is required for energy homeostasis (19).

However, we have a poor understanding of the requirements and microbial synthesis of B vitamins across vertebrate groups. It is thought that omnivores and carnivores obtain sufficient B vitamins through dietary input, especially animal matter (2). However, given that B vitamins are water soluble, animals cannot store much within the body and rely on a constant supply of these cofactors. Therefore, omnivores and carnivores may rely on microbial synthesis of B vitamins during certain times of low prey availability. Further, some vitamins are present in low concentrations or absent in plant-based diets and so may represent significant nutritional challenges for herbivores. For example, vitamin $B_{12}$ (cobalamin) is not found in plants, and so herbivores rely heavily on gut microbes for its production and acquisition (21). Vitamin deficiencies in wildlife are rarely detected due to the difficulty of studying wild animals and the fact that clinical signs of vitamin deficiencies, such as hemorrhaging throughout the body and neurological effects, are complex and not acute until later, life-threatening stages (2). However, vitamin deficiencies have been detected in numerous species of captive zoo animals (22–24). Therefore, understanding the microbial contributions to these processes could be important for captive breeding and conservation practices (25).

The links between gut bacterial community structures and functions in maintaining vitamin homeostasis have not been investigated across mammalian groups (12). Earlier culture-based studies isolated vitamin-producing microbes from the guts of various animals (26). However, this early work was largely focused on laboratory rodents or ruminant herbivores of agricultural interest, such as cattle and sheep (27). Recent advances in DNA-sequencing technology have allowed for more thorough investigation of microbial functions. In humans, there are several bacterial genera that are known for their ability to synthesize B vitamins, such as *Clostridium*, which is associated with the

synthesis of folate, cobalamin, niacin, and thiamine (14); *Bifidobacterium* is associated with folate synthesis (28), and *Bacteroides* is associated with the production of riboflavin, niacin, pantothenate, and pyridoxine (14). Considering that B-vitamins biosynthesis is prevalent in the human gut, a recent study analyzed the sequences of 256 human gut microbes for their ability to synthesize eight B vitamins (14). In humans, it seems that the synthesis of B vitamins occurs through the cooperation of numerous microbial taxa (14). However, despite roughly 80 years of interest in microbial contributions to vitamin homeostasis, our understanding of these processes across mammalian hosts with different feeding strategies is still severely limited.

In this study, we analyze the gut microbial communities from a diverse set of mammalian species with various feeding strategies to understand how microbes contribute to the synthesis of B vitamins. We utilize previously published metagenomic data sets that were generated from shotgun sequencing of fecal DNA from 39 mammalian species (11). These species can be divided into three feeding strategies: 7 carnivores, 11 omnivores, and 21 herbivores. Across feeding strategies, we compare the relative abundances of genes associated with vitamin B synthesis, the community structures of vitamin-synthesizing microbes, and the relative contributions of different microbial taxa that perform these functions. We predicted that herbivorous mammals would have higher abundances of genes associated with vitamin synthesis, given that plant material is often low in B vitamins (21), and it is thought that microbial vitamin synthesis is an adaption for herbivorous animals. Additionally, given the differences in microbial community structure between foregut- and hindgut-fermenting herbivores (10), we also compare aspects of vitamin synthesis between these groups, though we did not have any specific predictions for this comparison. Overall, our work offers the first comprehensive assessment of the communities that synthesize B vitamins in wild mammals. This information will broaden our understanding of host-microbe interactions and could have important applications for zoo husbandry and wildlife conservation.

## RESULTS

**Relative abundances of vitamin synthesis genes.** We first compared the relative abundances of genes associated with B-vitamin synthesis by using the publicly available sequences and the program MG-RAST (29, 30). We selected sequences that were identified as being associated with B-vitamin synthesis according to the Subsystems PubSEED classification system (31, 32) and calculated the relative abundance of those reads in relation to the number of total reads.

We identified two B vitamins where the abundances of microbial pathways associated with their synthesis varied significantly across mammalian feeding strategies. First, we observed a significant difference in the abundances of genes associated with cobalamin ($B_{12}$) synthesis across groups (Fig. 1; $P = 0.035$). Specifically, carnivores exhibited roughly $1.5\times$-higher abundances of these genes compared to other feeding groups. We also found significant differences in the relative abundances of genes associated with thiamine synthesis (Fig. 1; $P = 0.0481$), such that carnivores exhibited roughly $1.5\times$-higher abundances compared to other mammals with other feeding strategies. We did not detect differences in the relative abundances of genes associated with the synthesis of the other remaining B vitamins across feeding groups (biotin, folate, niacin, pantothenate, pyridoxine, and riboflavin) (Fig. 1; $P > 0.05$ for all).

**Community structure of vitamin-synthesizing microbes.** Next, we compared the taxonomic compositions of bacterial communities contributing to B-vitamin synthesis across mammalian feeding strategies. We accomplished this by isolating the sequences identified as B-vitamin synthesis genes, determining the taxonomic composition of these sequences using the Reference Sequence (RefSeq) database (33), and then comparing the community structures using Bray-Curtis dissimilarity metric in QIIME2 (34). Using principal-coordinate analysis (PCoA), we saw that the taxonomic community structures of vitamin synthesis genes were distinct across mammalian feeding strategies (Fig. 2). These visual differences were supported statistically, as permutational multivariate analysis of variance (PERMANOVA) revealed significant differences across

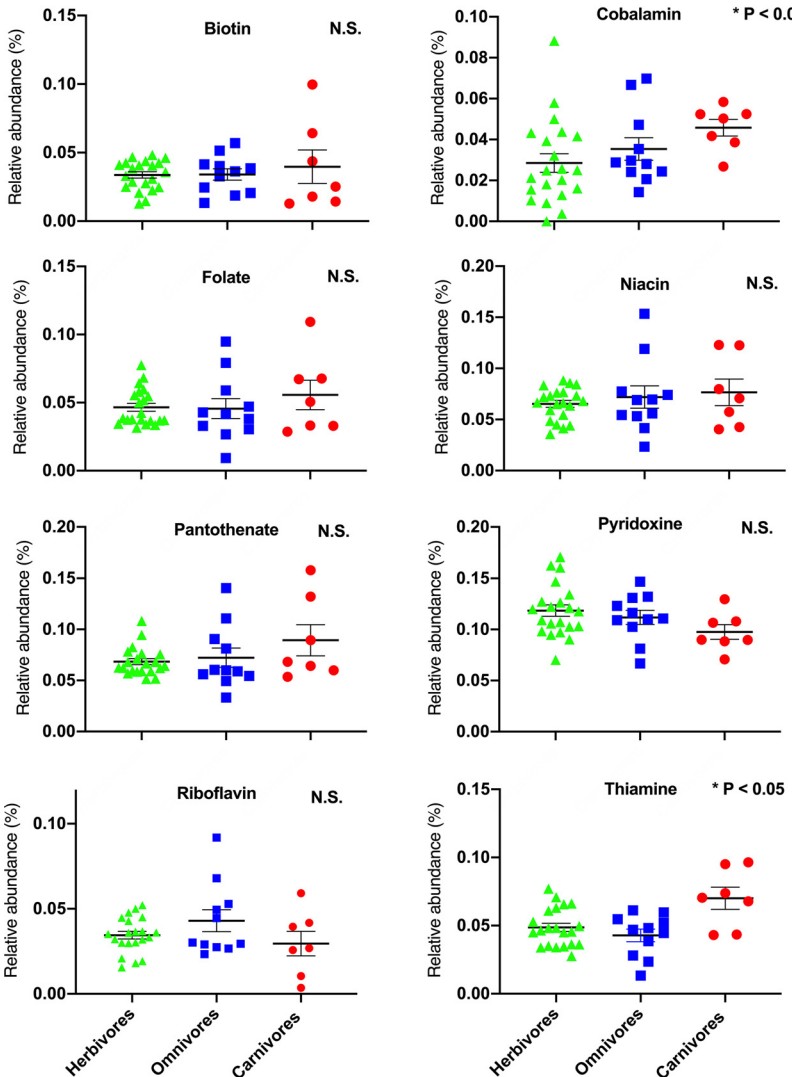

**FIG 1** Relative abundances of genes associated with the synthesis of various B vitamins across mammalian feeding groups. Kruskal-Wallis one-way ANOVA was used to compare groups with different feeding strategies. N.S., not significant. Lines represent mean ± standard error. Groups are denoted as follows: herbivores, green triangles; omnivores, blue squares; carnivores, red circles.

feeding strategies for all 8 B vitamins (Table 1). For each vitamin, we conducted pairwise comparisons and found that the taxonomic community structure differed between herbivorous and carnivorous mammals for all 8 vitamins (Table 1). Similarly, herbivores differed significantly from omnivores for all vitamins except for cobalamin (corrected $P$ value > 0.05). Pairwise comparisons did not detect any significant differences in microbiome community structures between omnivorous and carnivorous mammals for any B vitamin. Thus, herbivores have distinct compositions of microbes that conduct vitamin synthesis compared to other feeding groups (Fig. 3).

**Taxonomic abundances in the community of vitamin-synthesizing microbes.** Next, we compared the relative abundances of specific bacterial genera that contributed to vitamin synthesis and compared these abundances across mammalian feeding strategies. For each B vitamin, the relative abundances of microbial genera (as a portion of the community associated with each particular B vitamin) were uploaded into the program STAMP (35), and relative abundances were compared using nonparametric Kruskal-Wallis H-tests with Games-Howell *post hoc* tests.

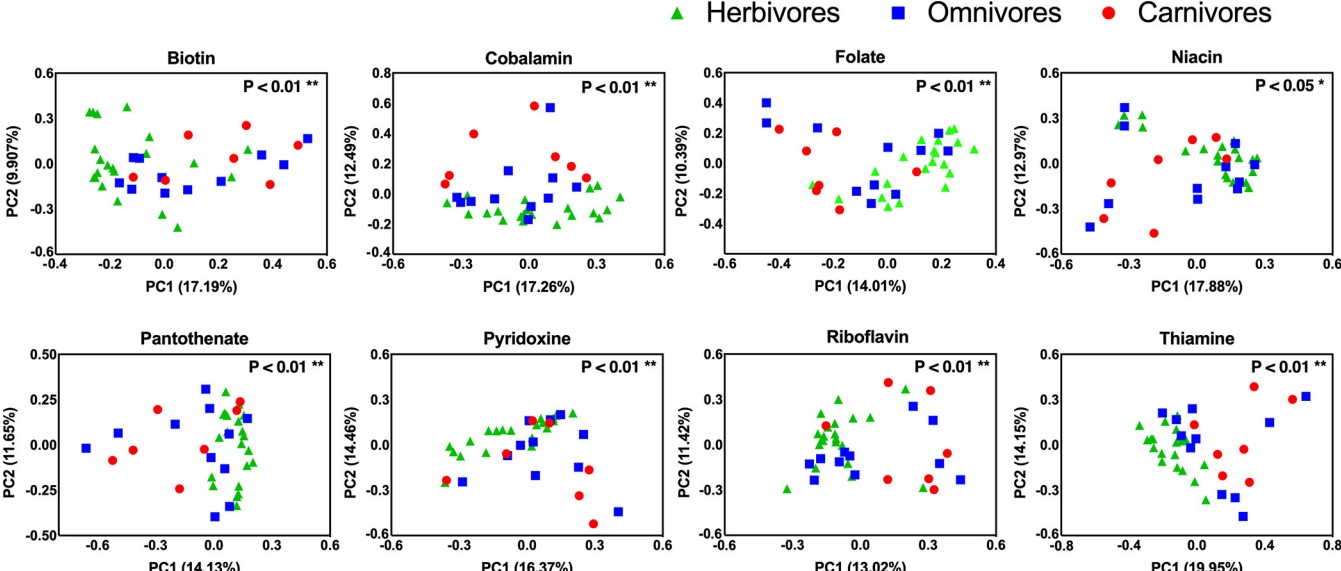

**FIG 2** Taxonomic community structures of metagenomic reads associated with vitamin synthesis. Principal-coordinate analysis (PCoA) plots were generated using Bray-Curtis dissimilarity metrics of taxon abundances of bacteria involved in vitamin synthesis. Differences in taxonomic community structure were statistically assessed by PERMANOVAs. Groups are denoted as follows: herbivores, green triangles; omnivores, blue squares; carnivores, red circles.

We identified many microbial taxa that contribute differentially to vitamin synthesis across mammalian groups. For example, the genus *Syntrophomonas* contributed to around 30% of the sequences associated with biotin synthesis in herbivorous mammals, whereas in omnivores and carnivores, *Syntrophomonas* contributed <15% of the sequences associated with biotin synthesis. Further, in carnivores, some species (echidna, bush dog, polar bear, and hyena) hosted *Acinetobacter* spp. that contributed to biotin synthesis, while this genus was not detected to contribute to biotin synthesis in herbivores or omnivores (Fig. 3). The genera *Desulfitobacterium* and *Eubacterium* contributed more to the synthesis of cobalamin in herbivores than in omnivores and carnivores. In Fig. 3, we highlight example microbial genera that are more enriched in herbivores and carnivores; additional taxa that varied across feeding strategies can be found in Fig. S1 in the supplemental material. Overall, these findings highlight those varying microbial taxa likely to contribute to vitamin synthesis across mammalian feeding strategies.

**Comparisons between foregut- and hindgut-fermenting herbivores.** Given that foregut- and hindgut-fermenting mammals have significantly different microbial community structures (10), we sought to compare the abundances of vitamin synthesis pathways, the bacterial community structures associated with these functions, and the contributions of microbial taxa to these function between these digestive strategies. Our data indicated that the relative abundances of genes associated with B-vitamin synthesis were not significantly different between these two groups (Fig. S2). We did find that foregut- and hindgut-fermenting mammals varied in the taxonomic community structure of microbes contributing to synthesis of niacin (PERMANOVA: $P = 0.003$) and riboflavin (PERMANOVA: $P = 0.037$) but not any other B vitamins (Fig. S3). Last, we identified several microbial taxa that contribute differentially to the synthesis of cobalamin, niacin, pantothenate, pyridoxine, and riboflavin between these digestive strategies (Fig. 4). For example, our data suggest that *Bacteroides* contributes significantly more to riboflavin synthesis for foregut fermenters ($P = 0.0001$). Further, *Mycobacterium* contributes more to niacin synthesis pathways in foregut fermenters than in hindgut fermenters ($P = 0.006$). Additional microbial genera that differentially contributed to vitamin synthesis between foregut- and hindgut-fermenting herbivores can be found in Fig. 4.

**TABLE 1** Results from total and pairwise PERMANOVA comparisons across mammalian feeding groups[a]

| Vitamin | Whole model | | Pairwise comparisons | | |
|---|---|---|---|---|---|
| | Pseudo-F | P | | Pseudo-F | Corrected P |
| Biotin | 2.17 | **0.002** | Herb vs Omni | 2.52 | **0.0045** |
| | | | Herb vs Carn | 2.59 | **0.0045** |
| | | | Omni vs Carn | 1.10 | 0.336 |
| Cobalamin | 2.19 | **0.001** | Herb vs Omni | 1.76 | 0.0525 |
| | | | Herb vs Carn | 3.21 | **0.003** |
| | | | Omni vs Carn | 1.40 | 0.0525 |
| Folate | 2.09 | **0.001** | Herb vs Omni | 1.75 | **0.0195** |
| | | | Herb vs Carn | 2.97 | **0.003** |
| | | | Omni vs Carn | 1.47 | 0.099 |
| Niacin | 1.81 | **0.01** | Herb vs Omni | 1.84 | **0.0405** |
| | | | Herb vs Carn | 2.18 | **0.024** |
| | | | Omni vs Carn | 1.25 | 0.197 |
| Pantothenate | 1.86 | **0.002** | Herb vs Omni | 1.05 | **0.006** |
| | | | Herb vs Carn | 2.29 | **0.006** |
| | | | Omni vs Carn | 2.29 | 0.411 |
| Pyridoxine | 1.85 | **0.003** | Herb vs Omni | 2.14 | **0.0135** |
| | | | Herb vs Carn | 2.30 | **0.0135** |
| | | | Omni vs Carn | 0.97 | 0.507 |
| Riboflavin | 1.74 | **0.004** | Herb vs Omni | 1.88 | **0.0105** |
| | | | Herb vs Carn | 2.09 | **0.0105** |
| | | | Omni vs Carn | 1.06 | 0.355 |
| Thiamine | 2.85 | **0.001** | Herb vs Omni | 3.00 | **0.0015** |
| | | | Herb vs Carn | 4.56 | **0.0015** |
| | | | Omni vs Carn | 0.94 | 0.497 |

[a]Abbreviations: Herb, herbivore; Omni, omnivore; Carn, carnivore. Boldface P values represent significance.

## DISCUSSION

In this study, we analyzed and compared the microbial communities that contribute to B-vitamin synthesis across 39 species of mammals. Our results showed the following: (i) the relative abundances of genes associated with cobalamin and thiamine synthesis were significantly enriched in carnivores, (ii) the community structures of microbes and contributions of specific microbial taxa to B-vitamin synthesis varied significantly based on host feeding strategy, and (iii) there were few differences between foregut- and hindgut-fermenting mammals in the abundance and composition of vitamin-synthesizing microbes. Collectively, our data provide insight into convergence of microbial functions across mammals with distinct feeding strategies and may have implications for animal husbandry and conservation.

Overall, our understanding of the kinetics of vitamin metabolism across wild animals is limited, but the subject has been studied in humans and domestic animals. Given the water-soluble nature of B vitamins, they are poorly stored within the body, and therefore, a relatively constant dietary supply may be needed by animals (2). For example, in humans, systemic stores of thiamine are exhausted within 18 days (36), and domestic cats and dogs can develop deficiencies in 1 to 2 weeks (37). While vitamin deficiencies are rarely detected in wildlife due to the difficulty of studying wild animals and the complexity of clinical symptoms (2), several case studies have detected vitamin deficiencies in captive zoo animals (22–24).

Interestingly, we did not find support for the hypothesis that herbivorous mammals would have higher abundances of genes associated with vitamin synthesis. Rather, we found that the abundances of genes associated with the synthesis of cobalamin and

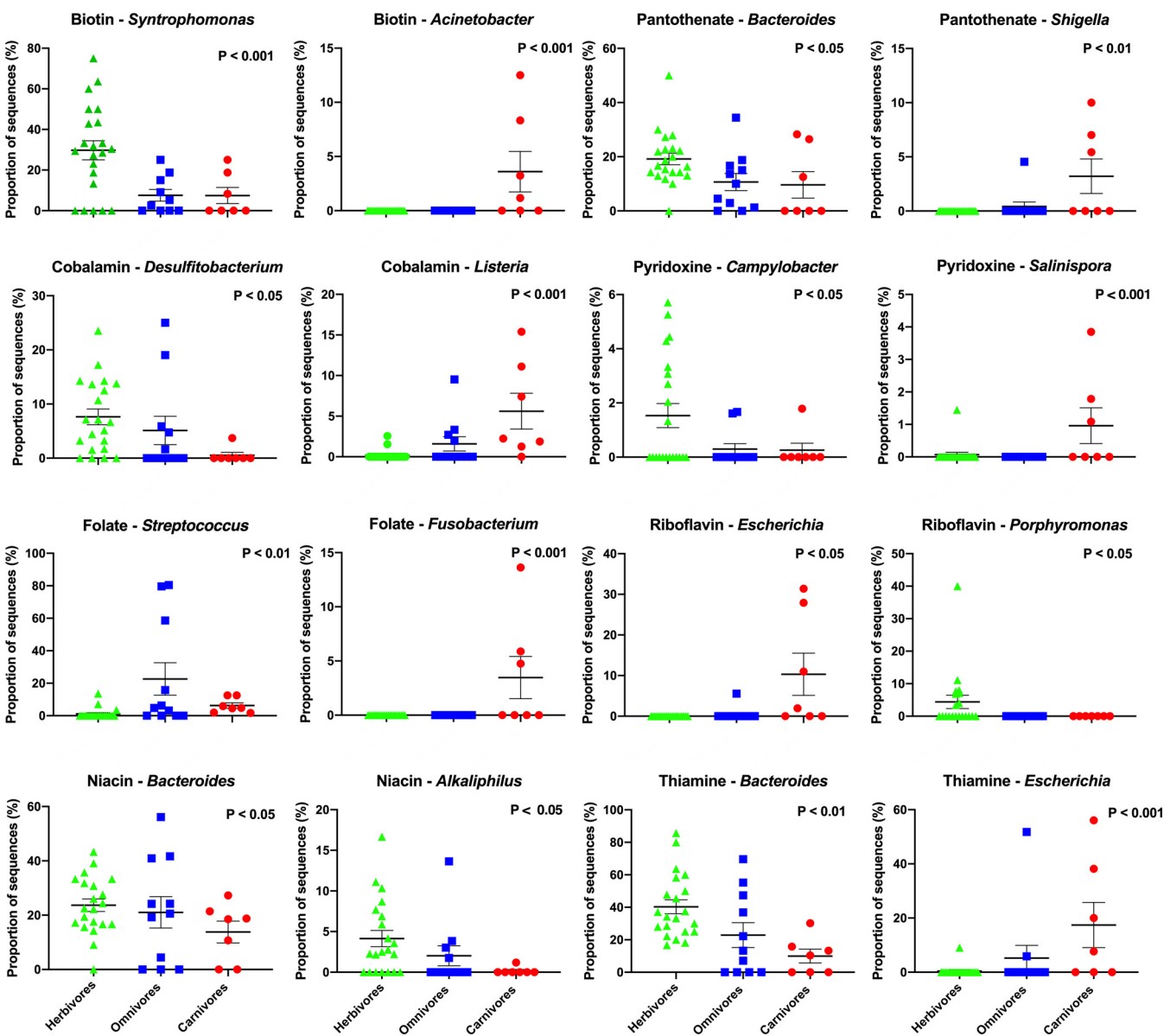

**FIG 3** The relative contributions of various bacterial genera to vitamin synthesis pathways, compared by mammalian feeding strategies. *P* values are determined by Kruskal-Wallis one-way analysis of variance (ANOVA) with the Games-Howell *post hoc* test. Lines represent mean ± standard error. Groups are denoted as follows: herbivores, green triangles; omnivores, blue squares; carnivores, red circles.

thiamine were higher in carnivorous mammals than in herbivores or omnivores. Both cobalamin and thiamine are important cofactors for enzymes involved in essential metabolic processes, and deficiencies can significantly alter central functions like growth, reproduction, and immunity (38, 39). Our finding of higher abundances of genes associated with the synthesis of these vitamins in carnivores is contrary to prior arguments that microbial synthesis of B vitamins might be especially important for herbivores, given the low abundance of these nutrients in plant material (2, 12, 27). However, the nutritional physiology and ecology of carnivores have been relatively overlooked compared to those of other feeding groups (40), and so we may have a poor understanding of vitamin metabolism and requirements in carnivorous animals. Again, while vitamin deficiencies are rarely detected in wildlife, they have been shown in captive animals. For example, thiamine deficiencies have been observed in captive cheetahs (22), lions (23), and Pacific harbor seals (24).

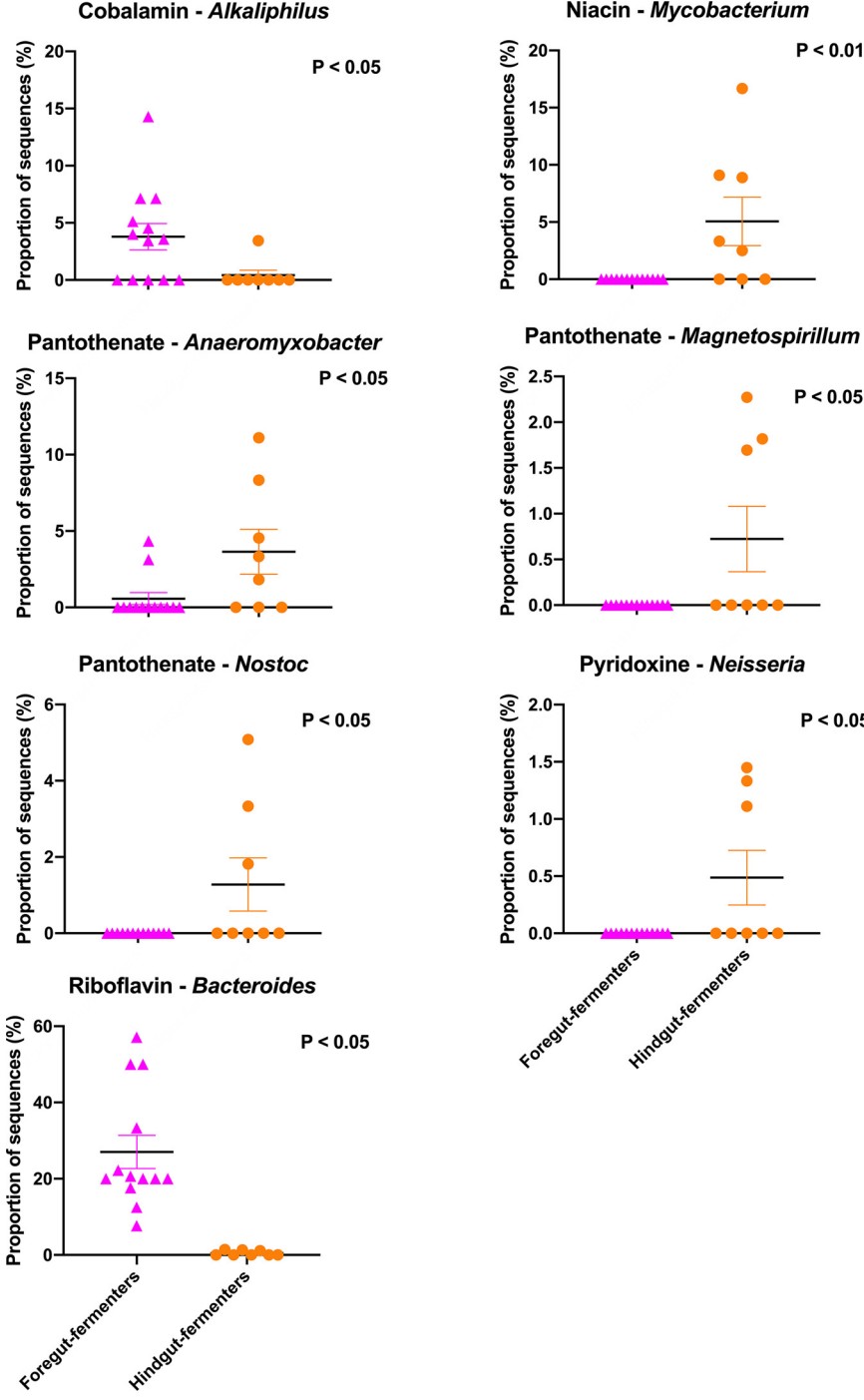

**FIG 4** The relative contributions of various bacterial genera to vitamin synthesis pathways, compared between foregut- and hindgut-fermenting herbivores. *P* values were determined by Kruskal-Wallis test with the Games-Howell *post hoc* test. Lines represent mean ± standard error. Groups are denoted as follows: foregut fermenters, pink triangles; hindgut fermenters, orange circles.

One central component of carnivore feeding ecology is the frequency of feeding, which can be dictated by prey availability, gut size, and other ecological factors. Most notably, feeding frequency scales negatively with body mass, such that large-bodied carnivores often experience prolonged intervals between meals (41). For example, the largest terrestrial carnivore, the polar bear (*Ursus maritimus*), feeds every 29 days on average but may experience more prolonged periods between meals. Another large-

mSystems®

bodied carnivore, the lion (*Panthera leo*), typically feeds every week but can experience periods of 16 days without food in the wild (42). In our own analysis, these two species were the mammalian carnivores with the largest body masses and also had the highest relative abundances of genes associated with thiamine synthesis. Due to poor storage of B vitamins, periods of food restriction, fasting, and starvation can induce vitamin deficiencies in humans (43, 44) and domestic cats (45), and similar patterns might occur in wild carnivores.

The gut microbiome has been shown to be involved in periods of fasting. For example, the community structure of the gut microbiome shifts in response to periods of fasting or feeding in wild animals (46, 47). Further, biomedical research shows that the gut microbiome improves survival during periods of starvation (48, 49), as germfree animals die sooner under periods of food deprivation while experiencing similar rates of body mass loss as conventional animals. While it has been demonstrated that gut microbes change aspects of systemic nutrient homeostasis under periods of starvation, such as inducing hosts to increase hepatic production of ketone bodies (48), other nutritional interactions could also be at play. An early study investigated the potential role of thiamine in prolonging survival during periods of starvation. Here, germfree mice injected with thiamine survived longer than germfree mice injected with water (49). However, these results were not statistically significant and came from an experiment with a small sample size per group ($n = 6$). Thus, more studies are needed to understand the role that microbes may play in the vitamin homeostasis of carnivorous animals, and perhaps during periods of nutrient deprivation.

Across mammals, diet and phylogeny appear to be major factors sculpting the taxonomic membership and functional profiles of gut microbial communities (10, 11, 50). Similarly, previous studies suggest some effects of host phylogeny on the membership of vitamin-synthesizing microbes. For example, *Bifidobacterium* spp. isolated from primates are able to synthesize folate, while *Bifidobacterium* from other mammalian hosts does not (51). While we do not have enough phylogenetic breadth of mammalian species to robustly investigate effects of evolutionary history (which we discuss more below), our culture-independent techniques demonstrate that the community structures of microbes that synthesize B vitamins vary significantly based on host diet. That is, a different collection of microbial members conducts the synthesis of B vitamins for herbivores than the collection of microbes that synthesizes B vitamins for omnivores and carnivores. Interestingly, recent studies suggest that the process of vitamin synthesis in the human gut involves collaboration and cross-feeding among microbial species (14, 52), and it would be interesting to investigate these processes across mammalian hosts.

We did not observe many differences between foregut- and hindgut-fermenting herbivores in the abundances of genes associated with vitamin synthesis or the composition of the vitamin-synthesizing community. Previous studies have found differences in the taxonomic composition of the gut microbiome based on these digestive strategies (10) but have not addressed functional differences based on gut anatomy. It could be that the selective pressures for vitamin homeostasis are relatively similar between foregut- and hindgut-fermenting herbivores. However, this data set was generated using fecal samples, which are generally representative of the hindgut community but are unlikely to represent the community residing in the foregut (53, 54). Thus, future studies could compare the vitamin-synthesizing microbes using actual gut contents to investigate differences across digestive strategies.

While our results provide an important first look into microbial vitamin synthesis across mammals, it is important to recognize several caveats and limitations of our approaches. First, we used preexisting data that had a slight taxonomic skew within some of the feeding categories. Of our omnivorous species, 8 out of 11 were from the order Primates, and of our carnivorous species, 5 out of 7 species were from the order Carnivora. Previous research shows that the taxonomic composition of the microbiome follows host phylogeny (10, 50). With this limited data set, Muegge et al. found that

mSystems®

less than 0.5% of microbial functions correlated with phylogeny and thus concluded that the functional capabilities of the microbiome are evolutionarily labile (11). Feeding strategies have evolved numerous times independently across many mammalian lineages (55), and so wider metagenomic sequencing across the phylogenetic tree will allow us to further verify our own findings and better investigate the evolution of microbial functions across mammalian hosts.

Our study is also limited by the computational nature of our approaches. Based on our current understanding, the genes encoding enzymes associated with B-vitamin synthesis are relatively well conserved across distantly related microbial taxa (14, 56). However, only ~32% of metagenomic reads from this published data set of mammalian hosts could be assigned to particular functions (11). This rate of annotation is common for metagenomic analysis, including the human gut microbiome (57). Thus, there is a large proportion of microbial genes that have not been characterized in terms of function or taxonomy, which represents a pressing challenge for the microbiome field (58). Last, the assignment of vitamin-synthesizing genes to particular microbial taxonomic groups used the RefSeq database (33), a widely used and well-curated database of annotated genomes. However, the use of short metagenomic reads does not necessarily account for horizontal gene transfer (HGT), which is especially common in the gut (59). Thus, while vitamin-synthesizing genes may have been assigned to come from a particular genus (e.g., *Syntrophomonas*), these genes may be in the context of another organism's genome. Several approaches are available for detecting incidents of HGT (60), though they require considerable bioinformatics expertise and are outside the scope of the current investigation. Last, independent of computational limitations, our results are based only on sequencing approaches that may not translate to actual functional differences. For example, recent work on the pea aphid (*Acyrthosiphon pisum*) and its symbiotic bacterium (*Buchnera aphidicola*) used both genomic predictions of microbial vitamin synthesis and extensive feeding trials lacking specific B vitamins on both symbiotic and aposymbiotic hosts (61). Researchers found little concordance between the genomic predictions and physiological results (61). Overall, as is the case with most microbiome research, increased molecular characterization of unique microbes, maintenance and regular updates of sequence databases, and the design of accessible bioinformatics pipelines will improve our ability to compare microbial functions across environments. Finally, in the end functional and physiological investigations should test the predictions made through sequenced-based studies.

Despite these caveats, our results improve our understanding of host-microbe interactions across mammalian hosts in regard to vitamin synthesis and could have implications for animal husbandry or conservation (25). Zoos and other establishments regularly house animals under captive conditions. Additionally, imperiled animal species are increasingly undergoing captive breeding programs in an effort to increase their population numbers. The publicly available data set used in our study was a combination of captive and wild mammals, with the original studies concluding that the effects of captivity were minimal (10, 11). However, samples from only 4 species (3 herbivores and one omnivore) were collected from the wild. More-focused studies have demonstrated that captivity significantly alters the gut microbiome of animals but that these changes are not uniform or consistent across host species (62, 63). For example, several species of foregut-fermenting herbivores show minimal differences between wild and captive counterparts. Conversely, primates and carnivores exhibit significant differences in the gut microbiome between these states (62). Given that some mammalian species can experience vitamin deficiencies in captivity, especially during early life (22–24), understanding the specific changes or microbial losses that mammalian species incur in captivity could be important for their conservation.

## MATERIALS AND METHODS

**Preparation and community representation.** For our study, we used previously published metagenomic data that demonstrated functional differences between herbivores, omnivores, and carnivores (11). This shotgun sequencing data (MG-RAST project ID: mgp116) was obtained from the MG-RAST metagenomic

analysis server (29, 30), where sequences had already been assigned to particular functions using the Subsystems PubSEED classification system which is integrated with MG-RAST version 4.0.3 (31, 32). Publicly available data for the 39 samples contained 2,163,286 total reads (mean = 55,469 $\pm$ 28,724 [standard deviation {SD}] per sample; 261 $\pm$ 83 nucleotides per read) (11). We used the following parameters to identify matches between the query sequence and conserved domains: E value of $10^{-5}$, 60% nucleotide identity, and a minimal alignment length of 15 bp (64–67). We filtered the data to specifically select for sequences assigned to the synthesis of particular B vitamins: biotin, cobalamin, folate, niacin, pantothenate, pyridoxine, riboflavin, and thiamine (Table 1). For all methods described below, we first compared data by feeding strategies (herbivore, omnivore, carnivore) and by gut anatomy within herbivores (foregut fermenters versus hindgut fermenters).

We first calculated the relative abundances of vitamin synthesis pathways for each B vitamin in each gut metagenome with the equation below. Relative abundances were statistically compared with the Kruskal-Wallis one-way analysis of variance (ANOVA), using a significance threshold of $P = 0.05$.

$$\% \text{ relative abundance} = \frac{\text{no. of reads assigned to vitamin synthesis subsystem}}{\text{no. of total reads}} \times 100$$

**Measuring microbial difference: beta diversity.** Next, we compared the community compositions of microbes involved in the synthesis of various B vitamins. Here, we used the reads previously identified as being associated with vitamin synthesis and used MG-RAST to identify these sequences taxonomically using the Reference Sequence (RefSeq) database (33). The taxonomic profiles of these reads (only those identified as vitamin synthesis pathways) were exported from MG-RAST as a BIOM table, and then the resulting BIOM table was uploaded into the program QIIME2 (34), where we computed the compositional dissimilarities using the Bray-Curtis index and performed pairwise PERMANOVA comparisons across groups (feeding strategies or gut morphologies). We generated principal-coordinate analysis (PCoA) plots to visually represent these communities and statistically compared community compositions of vitamin-producing microbes by performing pairwise PERMANOVA comparisons across groups (feeding strategies or gut morphologies), with a significance threshold of $P = 0.05$. The $P$ values for pairwise comparisons were corrected using a Benjamini-Hochberg correction.

**Identifying microbial taxa that contribute to vitamin synthesis.** Last, we compared the relative abundances of microbial taxa that contribute to vitamin synthesis. Here, we calculated the relative taxonomic abundances as a proportion of the reads that had been identified as being associated with B-vitamin synthesis. For example, in the giraffe ~50% of the reads identified as being associated with biotin synthesis were identified as coming from the genus *Syntrophomonas*. The vitamin-specific relative abundances of taxa were uploaded into the statistical analysis of taxonomic and functional profiles software, STAMP (35), and compared using nonparametric Kruskal-Wallis H-tests with the Games-Howell *post hoc* test, all with significance thresholds of $P = 0.05$. Given that microbial members often collaborate in the production of B vitamins (14), and the limited sequencing depth of this current data set, microbial taxa were identified as "vitamin producers" if they contained any gene associated with a particular B vitamin (that is, they did not need to contain the entire set of enzymes required for producing a given B vitamin).

## SUPPLEMENTAL MATERIAL

Supplemental material is available online only.

**FIG S1**, PDF file, 0.1 MB.

**FIG S2**, PDF file, 0.1 MB.

**FIG S3**, PDF file, 0.1 MB.

## ACKNOWLEDGMENTS

We thank Nicholas Barts and Elizabeth Rudzki for comments that helped to improve the manuscript.

This research received no specific grant from any funding agency in the public, commercial, or not-for-profit sectors.

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
