## [Reviewer comments · mSystems]

Abundance and compositions of B-vitamin-producing microbes in the mammalian gut vary based on feeding strategies

Hisham Alrubaye and Kevin Kohl

Corresponding Author(s): Kevin Kohl, University of Pittsburgh

Review Timeline:

Submission Date:	March 15, 2021
Editorial Decision:	May 7, 2021
Revision Received:	July 22, 2021
Editorial Decision:	August 6, 2021
Revision Received:	August 10, 2021
Accepted:	August 11, 2021

Editor: Suzanne Ishaq

Reviewer(s): The reviewers have opted to remain anonymous.

Transaction Report:

DOI: <https://doi.org/10.1128/mSystems.00313-21>

May 7, 2021

Dr. Kevin Kohl
University of Pittsburgh
4249 Fifth Ave.
Pittsburgh, PA 15260

Re: mSystems00313-21 (Abundance and compositions of B-vitamin-producing microbes in the mammalian gut vary based on feeding strategies)

Dear Dr. Kevin Kohl:

Thank you for submitting your manuscript to mSystems. We have completed our review and I am pleased to inform you that, in principle, we expect to accept it for publication in mSystems. However, acceptance will not be final until you have adequately addressed the reviewer comments.

The reviewers and I agree that this is an interesting piece, and is well composed. Some additional detail is requested by the reviewers to clarify several points.

The authors clearly discuss the limitations of their study and offer directions for future work, and while this work would be bolstered by additional *in vivo* experimentation, I think the Discussion clearly highlights the difficulty in studying these mechanisms *in vivo* with a great deal of resolution, necessitating the approach taken by the authors.

Thank you for the privilege of reviewing your work. Below you will find instructions from the mSystemseitorial office and comments generated during the review.

Preparing Revision Guidelines

For complete guidelines on revision requirements, please see the Instructions to Authors at [link to page]. **Submissions of a paper that does not conform to mSystems guidelines will delay acceptance of your manuscript.**

Due to the SARS-CoV-2 pandemic, our typical 60 day deadline for revisions will not be applied. I hope that you will be able to submit a revised manuscript soon, but want to reassure you that the journal will be flexible in terms of timing, particularly if experimental revisions are needed. When you are ready to resubmit, please know that our staff and Editors are working remotely and handling submissions without delay. If you do not wish to modify the manuscript and prefer to submit it to another journal, please notify me of your decision immediately so that the manuscript may be formally withdrawn from consideration by mSystems.

Sincerely,

Suzanne Ishaq

Editor, mSystems

Journals Department
Reviewer comments:

Reviewer #1 (Comments for the Author):

This is a descriptive analysis of vitamin B production potential by the microbiome across 39 species of mammals using the Muegge et al. dataset. As the authors state, vitamin synthesis has received relatively little attention in microbiome studies, particularly those that compare large numbers of host species. Vitamin B is of particular interest given that it cannot be effectively stored by the body and that it can be produced by microbes and absorbed by the colon. All in all, I think this paper is straightforward and clearly written. The analyses are simple yet appropriate. As a result, my specific suggestions are relatively limited (see below). My main concern is that despite a dearth of data on vitamin production by the microbiome, the findings seem somewhat incremental due some key gaps in the dataset and the approach (which are admittedly hard to address without designing and executing a study to generate new data).

Specific comments:

Line 131: My understanding is that B vitamins are one of the few things that can be absorbed in the colon - that might be worth mentioning somewhere in the introduction.

Line 323 (and 363): How conserved are B vitamin genes - how much do we know about vitamin B synthesis pathways? How well does PubSEED characterize these pathways based on what we know? It seems like this information is critical for understanding the extent to which we should be confident in these results, but I don't see it described in detail anywhere.

Line 347: Muegge et al. has quite a few captive individuals as the authors note. I suggest reminding readers what proportion are captive (and also reminding readers of the actual species included so that they do not have to go back to Muegge). Depending on sample sizes, it might be useful to do statistics on the effect of captivity, either overall or within diet categories. I mention this because, while I think the argument about carnivorous animal feeding patterns is compelling, it also occurs to me that herbivores in captivity eat a quite distinct diet compared to wild herbivores. Specifically, I would expect lower risk of vitamin B deficiency in captive animals, perhaps as a result of supplements, but also because most herbivores are not eating leaves in captivity, but rather some sort of chow. Therefore, they are not experiencing the same dietary conditions as they would in the wild, and our expectations about the importance microbial contributions to vitamin B might shift (although perhaps the same could be said for carnivores). Information about how many individuals from each diet category were captive versus not could help parse this information, even without formal statistical testing.

Reviewer #3 (Comments for the Author):

The study from Alrubaye and Kohl entitled "Abundance and compositions of B-vitamin-producing microbes in the mammalian gut vary based on feeding strategies" is a well-written paper describing a series of computational analyses to determine the distribution and abundance of B-vitamin synthesis across mammalian herbivores, carnivores or omnivores. This paper is of interest to the field as it provides insight into the potential importance of gut bacteria in the vitamin homeostasis of various mammals. The paper is technically sound, and the overall quality of this work is good. The overall writing is comprehensive and insightful with minor issues in the Methods section, in which the criterion/parameters to determine a "B-vitamin producer" is not clearly stated.

My only major remark is that this study is based on genomic prediction of vitamin synthesis and might not translate to actual functional differences. This aspect is already addressed by the authors, together with other limitations of the study, in the discussion, and I am pleased with that.

Some minor issues:

L193: please cite (Fig. 3).

L353-354: Please add the NCBI BioProject Record.

L358: please specify which genes were used to consider that a pathway for a specific B-vitamin was present in each vitamin producer. Was a "positive" B-vitamin producer required to possess all the genes involved in the biosynthesis of a specific vitamin or just few of them? Given that you used reads, what was the cutoff for the coverage of a specific gene in order to be considered present?

We thank the reviewers for their thoughtful comments that strengthened our manuscript. We have responded to each comment below with our response in bold italics and preceded by (---).

Reviewer #1 (Comments for the Author):

This is a descriptive analysis of vitamin B production potential by the microbiome across 39 species of mammals using the Muegge et al. dataset. As the authors state, vitamin synthesis has received relatively little attention in microbiome studies, particularly those that compare large numbers of host species. Vitamin B is of particular interest given that it cannot be effectively stored by the body and that it can be produced by microbes and absorbed by the colon. All in all, I think this paper is straightforward and clearly written. The analyses are simple yet appropriate. As a result, my specific suggestions are relatively limited (see below). My main concern is that despite a dearth of data on vitamin production by the microbiome, the findings seem somewhat incremental due some key gaps in the dataset and the approach (which are admittedly hard to address without designing and executing a study to generate new data).

--- We appreciate the reviewer's comments here. We agree that experiments will be required to fully test the ideas presented in our paper, and hope that this data set will spur those future studies.

Specific comments:

Line 131: My understanding is that B vitamins are one of the few things that can be absorbed in the colon - that might be worth mentioning somewhere in the introduction.

--- *This information is certainly helpful for the reader, and so we now write: "Here we specifically focus on B-vitamins, which can be divided into 8 types (biotin, cobalamin, folate, niacin, pantothenate, pyridoxine, riboflavin and thiamin), all of which can be synthesized by gut microbes (2, 15) and absorbed in the hindgut (16)."*

Line 323 (and 363): How conserved are B vitamin genes - how much do we know about vitamin B synthesis pathways? How well does PubSEED characterize these pathways based on what we know? It seems like this information is critical for understanding the extent to which we should be confident in these results, but I don't see it described in detail anywhere.

--- While the literature on the conservation vitamin B genes is limited, a recent publication (Das, et al., 2019), focused on vitamin metabolism of human gut microbiota and highlighted that across distantly related microbial species the genes encoding for the enzymes involved in the synthesis of riboflavin are well conserved. Another study (Magnúsdóttir et al. 2015) showed that the vitamin B genes are relatively conserved across 256 human gut microbiota organisms. PubSEED supports access to one of the largest collection of genomes from the academic community and provides consistent and accurate annotations and analysis for sequences. We have included the information in the text by writing: "Based on our current understanding, the genes encoding enzymes associated with B-vitamin synthesis are relatively well-conserved across distantly related microbial taxa (14, 56)."

Line 347: Muegge et al. has quite a few captive individuals as the authors note. I suggest reminding readers what proportion are captive (and also reminding readers of the actual species included so that they do not have to go back to Muegge). Depending on sample sizes, it might be useful to do statistics on the effect of captivity, either overall or within diet categories. I mention this because, while I think the argument about carnivorous animal feeding patterns is

compelling, it also occurs to me that herbivores in captivity eat a quite distinct diet compared to wild herbivores. Specifically, I would expect lower risk of vitamin B deficiency in captive animals, perhaps as a result of supplements, but also because most herbivores are not eating leaves in captivity, but rather some sort of chow. Therefore, they are not experiencing the same dietary conditions as they would in the wild, and our expectations about the importance microbial contributions to vitamin B might shift (although perhaps the same could be said for carnivores). Information about how many individuals from each diet category were captive versus not could help parse this information, even without formal statistical testing.

--- Unfortunately, there are only 4 samples from wild individuals, thus limiting the ability to compare these statistically. We now write: “The publicly available dataset used in our study was a combination of captive and wild mammals, with the original studies concluding that the effects of captivity were minimal (10, 11). However, samples from only 4 species (3 herbivores and one omnivore) were collected from the wild. More focused studies have demonstrated that captivity significantly alters the gut microbiome of animals, but that these changes are not uniform or consistent across host species (60, 61).”

Reviewer #3 (Comments for the Author):

The study from Alrubaye and Kohl entitled "Abundance and compositions of B-vitamin-producing microbes in the mammalian gut vary based on feeding strategies" is a well-written paper describing a series of computational analyses to determine the distribution and abundance of B-vitamin synthesis across mammalian herbivores, carnivores or omnivores. This paper is of interest to the field as it provides insight into the potential importance of gut bacteria in the vitamin homeostasis of various mammals. The paper is technically sound, and the overall quality of this work is good. The overall writing is comprehensive and insightful with minor issues in the Methods section, in which the criterion/parameters to determine a "B-vitamin producer" is not clearly stated.

My only major remark is that this study is based on genomic prediction of vitamin synthesis and might not translate to actual functional differences. This aspect is already addressed by the authors, together with other limitations of the study, in the discussion, and I am pleased with that.

---We thank the reviewer for their comments to improve our manuscript. We fully agree on the limitations of our data, but are glad that we have addressed these ideas sufficiently in the previous draft. Below we outline our specific changes to address the criteria for microbial taxa to be identified as “B-vitamin producers”.

Some minor issues:

L193: please cite (Fig. 3).

--- This change has been made.

L353-354: Please add the NCBI BioProject Record.

--- We have added the NCBI BioProject record ((NCBI Sequence Read Archive: SRA030940) to the manuscript.

L358: please specify which genes were used to consider that a pathway for a specific B-vitamin was present in each vitamin producer. Was a "positive" B-vitamin producer required to possess all the genes involved in the biosynthesis of a specific vitamin or just few of them? Given that

you used reads, what was the cutoff for the coverage of a specific gene in order to be considered present?

--- We appreciate this comment by the reviewer, and explain our approach in the methods by now writing: “Given that microbial members often collaborate in the production of B-vitamins (14), and the limited sequencing depth of this current dataset, microbial taxa were identified as “vitamin producers” if they contained any gene associated with a particular B-vitamin (that is, they did not need to contain the entire set of enzymes required for producing a given B-vitamin).”

• Additional details for cutoffs can now be found in the Methods, where we now write: “Publicly available data for the 39 samples contained 2,163,286 total reads (mean = $55,469 \pm 28,724$ (SD) per sample; 261 ± 83 nucleotides per read) (11). We used the following parameters to identify high-confidence matches between the query sequence and conserved domains: e-value of 10^{-5} , 60% identity, and a minimal alignment length of 15 amino acids.”

August 6, 2021

Dr. Kevin Kohl
University of Pittsburgh
4249 Fifth Ave.
Pittsburgh, PA 15260

Re: mSystems00313-21R1 (Abundance and compositions of B-vitamin-producing microbes in the mammalian gut vary based on feeding strategies)

Dear Dr. Kevin Kohl:

Thank you for submitting your manuscript to mSystems. We have completed our review and I am pleased to inform you that, in principle, we expect to accept it for publication in mSystems. However, acceptance will not be final until you have adequately addressed the reviewer comments.

The authors have done well to address reviewer comments, and the reviewers and I agree that these modifications improved the paper. The reviewers and I agree that the manuscript is acceptable for publication, however, reviewer #2 noted a few places where the methods could be more specific to aid in reproducibility. Likely the SRA is still under embargo, but the other two comments appear easily addressed. Once these are corrected, I do not see the need for an additional round of review, and I look forward to formally accepting your manuscript.

Preparing Revision Guidelines

For complete guidelines on revision requirements for your article type, please see the journal Article Types requirement at <https://journals.asm.org/journal/mSystems/article-types>. **Submissions of a paper that does not conform to mSystems guidelines will delay acceptance of your manuscript.**

Sincerely,

Suzanne Ishaq

Editor, mSystems

Journals Department
Reviewer comments:

Reviewer #1 (Comments for the Author):

All of my comments have been addressed. I have to admit I am now curious how the results change if only entire pathways (in single microbial taxa) are included, but I don't think this analysis is necessary for publication. Again, we know so little about these microbes and pathways that the data presented represent a good start for piquing more interest.

Reviewer #3 (Comments for the Author):

I would like to thank the authors for clarifying and addressing the comments received.

I just would encourage the authors to double check the NCBI BioProject record, given that the identifier added to the manuscript (SRA030940) gives no hit when searched on NCBI, MG-RAST and on the ENA website.

In addition, it's still unclear which software the authors used to go from reads to amino acids (see text at lines 371-373). Also, are these "conserved domains" defined in any previous paper? Alternatively, please explain how this analysis was conducted (please add the appropriate references).

Why did you use the values described at lines 372-373? Could you refer to any previous publication? "High confidence" does not sound like the appropriate term here, unless you use a much higher identity (>85% amino acid identity). Please specify if you used 60% amino acid or 60% nucleotide identity. It's unclear why you write "amino acids" here, given that your description at lines

155-169 refers to genes and there is no mention to proteins.

We thank each reviewer for their thoughtful comments that strengthened our manuscript. We have responded to each comment below with our response in bold italics and preceded by (---).

Reviewer #1 (Comments for the Author):

All of my comments have been addressed. I have to admit I am now curious how the results change if only entire pathways (in single microbial taxa) are included, but I don't think this analysis is necessary for publication. Again, we know so little about these microbes and pathways that the data presented represent a good start for piquing more interest.

--- We appreciate the time that the reviewer has put forward to provide us with comment to improve our manuscript.

Reviewer #3 (Comments for the Author):

I would like to thank the authors for clarifying and addressing the comments received.

I just would encourage the authors to double check the NCBI BioProject record, given that the identifier added to the manuscript (SRA030940) gives no hit when searched on NCBI, MG-RAST and on the ENA website.

--- We appreciate this comment by the reviewer, and we apologize for the inconvenience. We are also perplexed that these data are not publicly available through NCBI, but as they were not our original sequencing data, we have no control over this (though we did submit a request to the SRA to have these data released). Though, we only ever accessed data through MG-RAST, and not through NCBI, so in the manuscript we now write that we used data from project ID: mgp116 on MG-RAST as the reference.

We now write at L57-361 "This shotgun sequencing data (MG-RAST project ID: mgp116) was obtained from MG-RAST metagenomic analysis server (1, 2), where sequences had already been assigned to particular functions using the Subsystems PubSEED classification system which is integrated with MG-RAST version 4.0.3 (3, 4)"

In addition, it's still unclear which software the authors used to go from reads to amino acids (see text at lines 371-373). Also, are these "conserved domains" defined in any previous paper? Alternatively, please explain how this analysis was conducted (please add the appropriate references).

--- We appreciate the reviewer's comments here and apologize for the error, as reads were never translated to amino acids. MG-RAST identifies the composition of a microbial

community from the shotgun metagenomic data using sequence similarities. Based on the sequence similarity search against multiple databases, functions are assigned to specific taxa, this feature is provided Subsystems PubSEED classification via MG-RAST (29).

We now write at L357-361: “This shotgun sequencing data (MG-RAST project ID: mgp116) was obtained from MG-RAST metagenomic analysis server (1, 2), where sequences had already been assigned to particular functions using the Subsystems PubSEED classification system which is integrated with MG-RAST version 4.0.3 (3, 4).”

Why did you use the values described at lines 372-373? Could you refer to any previous publication? "High confidence" does not sound like the appropriate term here, unless you use a much higher identity (>85% amino acid identity). Please specify if you used 60% amino acid or 60% nucleotide identity. It's unclear why you write "amino acids" here, given that your description at lines 155-169 refers to genes and there is no mention to proteins.

--- We appreciate the reviewer’s comments here and will edit this error in the manuscript. The Assignment of functional classification into SEED database categories was performed in MG-RAST using the following parameters: a maximum e-value of 1e-5, a minimum nucleotide identity of 60% and a minimum alignment length of 15 bp. These parameters used on MG-RAST maximize sensitivity (65). We chose our parameters based on previous publications that characterized microbial communities (64-67).

We now write at L362-365: “ We used the following parameters to identify matches between the query sequence and conserved domains: e-value of 10^{-5} , 60% nucleotide identity, and a minimal alignment length of 15 base pairs (64–67).”

August 11, 2021

Dr. Kevin Kohl
University of Pittsburgh
4249 Fifth Ave.
Pittsburgh, PA 15260

Re: mSystems00313-21R2 (Abundance and compositions of B-vitamin-producing microbes in the mammalian gut vary based on feeding strategies)

Dear Dr. Kevin Kohl:

Your manuscript has been accepted, and I am forwarding it to the ASM Journals Department for publication. For your reference, ASM Journals' address is given below. Before it can be scheduled for publication, your manuscript will be checked by the mSystems senior production editor, Ellie Ghatineh, to make sure that all elements meet the technical requirements for publication. She will contact you if anything needs to be revised before copyediting and production can begin. Otherwise, you will be notified when your proofs are ready to be viewed.

As an open-access publication, mSystems receives no financial support from paid subscriptions and depends on authors' prompt payment of publication fees as soon as their articles are accepted. =

Publication Fees:

We recognize that the video files can become quite large, and so to avoid quality loss ASM suggests sending the video file via <https://www.wetransfer.com/>. When you have a final version of the video and the still ready to share, please send it to Ellie Ghatineh at eghatineh@asmusa.org.

Sincerely,

Suzanne Ishaq
Editor, mSystems

Journals Department
Fig. S2: Accept
Fig. S3: Accept
Fig. S1: Accept